# 3-Dimensional Porous Carbon with High Nitrogen Content Obtained from Longan Shell and Its Excellent Performance for Aqueous and All-Solid-State Supercapacitors

**DOI:** 10.3390/nano10040808

**Published:** 2020-04-23

**Authors:** Yuhao Liu, Xiaoxiao Qu, Guangxu Huang, Baolin Xing, Fengmei Zhang, Binbin Li, Chuanxiang Zhang, Yijun Cao

**Affiliations:** 1College of Chemistry and Chemical Engineering, Henan Polytechnic University, Jiaozuo 454000, China; LYH1441518335@163.com (Y.L.); hgxu@hpu.edu.cn (G.H.); baolinxing@hpu.edu.cn (B.X.); 18336815354@163.com (F.Z.); lbb4017@163.com (B.L.); 2Henan Key Laboratory of Coal Green Conversion, Henan Polytechnic University, Jiaozuo 454000, China; 3Collaborative Innovation Center of Coal Work Safety, Henan Polytechnic University, Jiaozuo 454000, China; 4College of nanoscience and nanotechnology, Pusan National University, Busan 46241, Korea; m18339169838@163.com; 5Henan Province Industrial Technology Research Institute of Resources and Materials, Zhengzhou University, Zhengzhou 450001, China; yijuncao@126.com

**Keywords:** 3-dimensional porous carbon, biomass, high nitrogen content, all-solid-state supercapacitors

## Abstract

Three-dimensional porous carbon is considered as an ideal electrode material for supercapacitors (SCs) applications owing to its good conductivity, developed pore structure, and excellent connectivity. Herein, using longan shell as precursor, 3-dimensional porous carbon with abundant and interconnected pores and moderate heteroatoms were obtained via simple carbonization and potassium hydroxide (KOH) activation treatment. The electrochemical performances of obtained 3-dimensional porous carbon were investigated as electrode materials in symmetric SCs with aqueous and solid electrolytes. The optimized material that is named after longan shell 3-dimensional porous carbon 800 (LSPC800) possesses high porosity (1.644 cm^3^ g^−1^) and N content (1.14 at %). In the three-electrode measurement, the LSPC800 displays an excellent capacitance value of 359 F g^−1^. Besides, the LSPC800 also achieves splendid specific capacitance (254 F g^−1^) in the two electrode system, while the fabricated SC employing 1 M Li_2_SO_4_ as electrolyte acquires ultrahigh power density (15930.38 W kg^−1^). Most importantly, LSPC800 electrodes are further applied into the SC adopting the KOH/polyvinyl alcohol (PVA) gel electrolyte, which reaches up to an outstanding capacitance of 313 F g^−1^ at 0.5 A g^−1^. In addition, for the all-solid-state SC, its rate capability at 50 A g^−1^ is 72.73% and retention at the 10,000th run is 93.64%. Evidently, this work is of great significance to the simple fabrication of 3-dimensional porous carbon and further opens up a way of improving the value-added utilization of biomass materials, as well as proving that the biomass porous carbons have immense potential for high-performance SCs application.

## 1. Introduction

With accelerative threats to the environment and excessive resource consumption, it is vitally necessary to seek clean, green, sustainable, and low-cost energy or energy storage technology [1,2]. In addition, the rapid society rhythm brings forth the necessity of developing efficient, miniaturized, and flexible devices for meeting the requirements of comfort and convenience [3]. At present, the widely used energy storage devices in actual life mainly include batteries and capacitors. The former possess superior energy density and long cycling lifetime, while the latter have high power density as well excellent safety [4]. However, both of them have the disadvantage of large bulk and less efficiency. Supercapacitors (SCs) seem to be the ideal alternative because they achieve both excellent energy density and power density compared with above two devices. Besides, SCs have also received great interest because of their reliable lifetime (~10^5^ cycles), excellent safety, high charging/discharging rate, and low cost [5,6,7].

In general, the electrochemical properties of SCs primarily rely on the electrode material. Typically, compared with some carbon materials, such as graphene-based materials, carbon fiber, and carbon black, the 3-dimensional porous carbon material has mostly been employed in the SCs because of the developed pore structure, high specific surface area, outstanding stability, mechanical strength, as well as light weight [8,9]. Therefore, the SCs assembled with 3-dimensional porous carbon material as electrodes obtain ideal capacitance and high capacitance retention. Zhang et al. [10] designed the oxygen-rich porous carbon from bituminous coal, which achieved splendid performance (270 F g^−1^ at 20 A g^−1^ high current density). Truc et al. [11] fabricated 3-dimensional carbon with a peculiar structure and displayed high capacitance (390 F g^−1^). Yang et al. [12] propose a novel strategy to develop the hierarchical hollow carbon sphere, achieving ultrahigh power density. Lee et al. [13] synthesized interconnected nitrogen-doped carbon material using graphite oxide (GO) and triblock copolymer (Pluronic P123) as precursors, which show excellent cycling property with outstanding capacitance retention (~98%) at the 5000th run. However, the synthesis of these porous carbon materials has some drawbacks, including a complex preparation process, expensive raw materials, and poor product controllability, which limit their widespread practice for SCs [3]. Thus, it is necessary to choose a suitable precursor to eliminate the above disadvantages.

Recently, biomass material has paid great attention in fabricating porous carbon owing to its excellent chemical nature, environmental friendliness, and controllable structure. In addition, biomass contains various heteroatoms, including oxygen, nitrogen, sulfur, and related surface functional groups. These heteroatoms can offer pseudocapacitance and improve the wettability in material surface. Some functional groups, such as N-Q, benefit the conductivity, contributing to electrochemical performances of biomass carbon material [14]. Furthermore, pore size distribution (PSD) is also crucial factor for biomass electrodes. Generally speaking, by using simple carbonization and KOH activation treatment, biomass porous carbon with high porosity and moderate PSD can be facilely synthesized. The high porosity, meaning large specific surface area (SSA), is conducive to offer plenty of active sites for ions and sufficient accommodation space for electrolytes. The moderate PSD, including abundant micropores and a certain number of mesopores, can provide rapid ion diffusion, which can greatly improve the capacitance retention performance [15]. Herein, biomass carbon is an ideal electrode material for SCs application. All kinds of biomass, including sesame husk [16], sugar cane bagasse [17], tree-bark [18], and even starch [19], were widely used to afford porous carbons for SCs application. Longan shell, as the precursor of biomass carbon, has the advantage of low cost, abundant sources, and loose structure, while the longan shell-based carbon possesses the excellent character of porous structure and moderate heteroatomic content [20]. Most importantly, longan shell-based carbon can be prepared in a large scale, which is favorable to the economic benefits. Therefore, longan shell-based carbon holds immense potential for SCs application.

Besides, the electrolyte is also considered as the essential impact affecting the electrochemical performance. In some studies about SCs, liquid electrolytes are employed owing to their good surface wettability and conductivity, but actual utilization of liquid electrolytes is unsatisfying because of its low ion selectivity, easy decomposition in high temperature, corroding electrodes, as well as inferior portability [21,22,23]. Solid electrolyte can not only solve the above problems, but is also employed as separator and media [24,25]. What is more, solid electrolyte can supply the outcomes of convenience, light weight, and environmental friendliness [26]. Because of the above advantages, solid-state SCs are widely used in miniature intelligent electronic equipment. Yu et al. [27] fabricated carbon materials with high porosity, thus the optimized sample gained excellent capability (265 F g^−1^) in solid-state SCs with the voltage window of 0–1.2 V. Patil et al. [28] integrated the nano-structure gold on an RGO (reduced graphene oxide)/ZnCo_2_O_4_ composite, so the prepared asymmetric solid SC delivers outstanding energy density (31 Wh kg^−1^). Wang et al. [29] synthesized nitrogen-riching activated carbon employing the urea as nitrogen source, which delivered an impressive capacitance value (306 F g^−1^) with gel electrolyte. Sarigamala et al. [30] reported the Ni-Co hydroxide and the hybrid SC device using solid electrolyte exhibits considerable energy density (35 Wh kg^−1^). Therefore, the solid electrolyte is vitally matching for actual SCs application.

In this work, 3-dimensional porous carbon was obtained using longan shell as precursor via simple carbonization and KOH activation treatments. By changing the activation temperature, detailed physicochemical and electrochemical properties of longan shell 3-dimensional porous carbon (LSPC) were studied. The fabricated LSPCs exhibit large SSA (reaching to 3089 m^2^ g^−1^), large porosity (maximum value of 1.644 m^3^ g^−1^), and moderate PSD. Thus, these LSPCs samples demonstrate outstanding properties (capacitance up to 359 F g^−1^) in three-electrode measurements using aqueous KOH, and achieve an ideal capacitance rate (up to 79.79% of initial value). Furthermore, the assembled SCs using LSPC800 as electrode in the 1 M Li_2_SO_4_ solution under the voltage range of 0–1.6 V also display an ideal capacitance value (254 F g^−1^) as well as ultrahigh power density (15930.38 W kg^−1^). Most, importantly, adopting LSPC800 electrodes fabricates all-solid-state SCs, which demonstrate the impressive specific capacitance (313 F g^−1^), outstanding rate performance (72.73% at 50 A g^−1^), and ideal cyclicity (93.64% capacitance retention in the 10,000th charging/discharging process).

## 2. Experimental

### 2.1. Materials and Reagents

The longan shell was collected from the market. Hydrochloric acid (HCl, 36–38 wt %) and potassium hydroxide (KOH) were provided by Yantai City Shuangshuang Chemical Reagent Co., Ltd. (Yantai, China). Lithium sulfate (Li_2_SO_4_) was bought from Tianjin City Kemiou Chemical Reagent Co., Ltd. (Tianjin, China). Polyvinyl alcohol (PVA) was obtained from Aladdin Industrial Corporation (Shanghai, China). Deionized water was applied in the whole experiment process and all chemical reagents were of analytical purity.

### 2.2. Preparation of Longan Shell 3-Dimensional Porous Carbons (LSPCs)

The longan shell 3-dimensional porous carbons were synthesized in a typical carbonization and KOH activation treatment, which is demonstrated in Figure 1. First of all, the longan shell was cleaned with water and ethanol to remove excess impurities. After remaining at 120 °C for 24 h, longan shell was ground into powder. In the carbonization process, the dried longan shell powder was heated to 600 °C for 1 h with a 5 °C min^−1^ heating rate under pure N_2_. The carbonized sample was denoted as C. Before the activation process, the 3 g C sample was added into the solution containing 9 g KOH with gentle magnetic stirring overnight, and afterwards, the obtained product was dried fully. The mixture sample was activated with the pure N_2_ to specific temperature, as well as remaining for 2 h at 5 °C min^−1^. After natural cooling to room temperature, the obtained sample was soaked in 50% HCl solution for wiping off redundant KOH and other impurities as well as washing using deionized water to neutral PH. Finally, the LSPCs were obtained after drying in the vacuum oven at 80 °C overnight. According to different activation temperatures, the products were named for LSPC*x*, where *x* is 700, 800, and 900, respectively.

### 2.3. Physicochemical Characterization

X-ray diffraction (XRD, Bruker D8 Advance, Karlsruhe, Germany) patterns were obtained by carrying Cu Kα radiation (*λ* = 1.54 Å) at 40 kV from 5° to 90° at the rate of 10° min^−1^. Raman spectroscopic technique (LabRAM HR Evolution, Tokyo, Japan) was used to characterize these samples. XPS (X-ray photoelectron spectroscopy) characterization was performed using Thermo ESCALAB 250XI spectrometer (Shanghai, China) under the pressure of 5 × 10^−11^ Torr. Scanning electron microscope (SEM, JSM-6390LV, Tokyo, Japan) and transmission electron microscope (TEM, JEM-2100, Tokyo, Japan) were used to evaluate the morphologies of samples. Nitrogen adsorption and desorption was carried out at 77 K on an Autosorb-iQ-MP (Quantachrome, Florida, USA). The BET (Brunauer-Emett-Teler) and DFT (Density functional theory) methods were adopted to obtain these parameters of SSA and PSD. The electrical conductivity of LSPCs was tested by measuring a four-probe power resistivity tester (FT-341A, Shanghai, China).

### 2.4. Electrode Preparation and Electrochemical Measurement

#### 2.4.1. Preparation of Electrode and Test in Three-Electrode System

The electrochemical measurements were performed on a CHI760E workstation (Chenhua co., LTD, Shanghai, China) with a conventional three-electrode system. The working electrode was fabricated by grinding LSPCs powder (80 wt %), conductive carbon black (10 wt %), and polytetrafluoroethylene binder (PTFE, 10 wt %) dispersed in ethanol. These mixtures were then cut into the disk with 10 mm diameter, and then integrated on the nickel foam. The active mass of a single working electrode is about 2~3 mg cm^−2^ after being treated in a vacuum oven at 100 °C. In addition, the Pt plate, Hg/HgO, and 6 M KOH were used as the counter electrode, reference electrode, and electrolyte, respectively. Galvanostatic charge–discharge (GCD) and cyclic voltammetry (CV) studies were conducted with the potential of −1 to 0 V, while the electrochemical impedance spectroscopy (EIS) measurements were performed in a frequency ranged from 100 kHz–0.01 Hz.

The specific capacitance (*C*, F g^−1^) was evaluated by discharging curves and is displayed as follows:(1)$$C=\frac{I\Delta t}{m\Delta V}$$
where *I* (A) and Δ*t* (s) represent the discharge current and time, respectively; Δ*V* (V) shows the voltage change (excluding the internal resistance (IR) drop) in the discharge process; and *m* (g) implies the mass of the active materials.

#### 2.4.2. Assembly and Test of Aqueous/All-Solid-State SCs

The aqueous SCs were fabricated in CR2016 button cell with LSPC800 electrode, and the polypropylene (PP) separator was soaked in 1 M Li_2_SO_4_ electrolyte. The all-solid-state SCs were assembled using PVA/KOH as electrolyte. PVA/KOH electrolyte was prepared using the traditional method. Firstly, 3 g of PVA is put into 20 mL of DI water then stirred in the 85 °C water bath until complete dissolution of PVA. Then, 6 mol L^−1^ KOH solution (15 mL) is put drop by drop at 90 °C to obtain a jell-like product, which is PVA/KOH gel. Afterwards, two working electrodes are submerged in PVA/KOH gel several times to ensure they are fully soaked and placed in the air to exclude extra moisture. Eventually, sandwich-like solid-state SC is assembled by compressing the pair of working electrodes and the current collector is nickel foam. The specific capacitance *C*_g_ (F g^−1^) is reckoned as follows:(2)$$C_{g}=\frac{2I\Delta t}{m\Delta V}$$
where *I* (A) and Δ*t* (s) represent the discharge current and time in GCD curves, respectively; Δ*V* (V) shows the voltage change (excluding the IR drop) in the discharge process; and *m* (g) implies the average mass of the active material in the two working electrodes.

The energy density (*E*, Wh kg^−1^) and power density (*P*, W kg^−1^) were calculated by GCD curve according to the following equations:(3)$$E=\frac{C_{g}{\left(\Delta V\right)}^{2}}{2\times 4\times 3.6}$$
(4)$$P=\frac{3600E}{\Delta t}$$

## 3. Results and Discussion

### 3.1. Physicochemical Characterization

XRD patterns are measured to determine the graphitization level, and are shown in Figure 2a. Two broad peaks located in ~23° and ~41° are observed in all LSPCs samples. The two characteristic peaks correspond with (002) and (100) diffractions, which proves the existence of graphite with relative low microcrystal. In addition, the two peaks are very weak, demonstrating the as-prepared samples are typical amorphous carbon materials [31]. The LSPCs are further characterized by Raman spectra in Figure 2b. Two typical peaks are at approximately 1300 and 1600 cm^−1^, which present the D and G band, respectively. The D peak is attributed to disorder and defects of the sp^3^–C effect, while the G peak is related to sp^2^–C, and the ratio of *I*_D_/*I*_G_ stands for the disorder degree of carbon material [32]. The obtained *I*_D_/*I*_G_ ratio values of LSPCs are 1.10–1.29. The high *I*_D_/*I*_G_ ratio values of LSPCs indicate a low graphitization level, as well as much disorder and defects in the structure, which is attributed to the influence of activated treatment. Furthermore, the *I*_D_/*I*_G_ ratio of LSPCs samples increases as the activation temperature rises. This can be explained in that the carbon in raw reacts with KOH and is etched gradually, resulting in high porosity and more defects [33].

As shown in Figure 3, XPS is measured to characterize elemental composition and their chemical environment of LSPCs. The full spectrum in Figure 3a indicates that LSPC800 mainly contains C, O, and N elements, and the atomic contents are exhibited in Table 1. It is obvious that, besides the dominant C element, all LSPCs also have a similar atomic content percentage. The heteroatoms are vitally essential for improving the surface properties of electrode materials [34]. In particular, nitrogen element and its relative functional groups can supply pseudo-capacitance and improve conductivity carbon materials [35]. The high resolution C 1s, O 1s, and N 1s spectra of LSPC800 is demonstrated in Figure 3b–d. From Figure 3b, there exist of four different types of functional groups at about 284.4, 285.6, 286.8, and 288.9 eV, which are consistent with sp^2^–C, sp^3^–C, C–O/C–O–C, and C=O groups, respectively [36]. The O 1s is also classified to three kinds of main functional groups. As shown in Figure 3c, these broad peak are O-I, O-II, and O-III, which can be attributed to C=O/C–OH (531.7 eV), C–O–C (532.8 eV), and chemisorbed O/H_2_O (535.5 eV), respectively. The O-containing functional groups can improve the wettability of the surface, and thus increase the chemical activity of electrodes [37]. In Figure 3d, the high resolution of N 1s contains two peaks, which are N-6 (398.3 eV) and N-Q (401.6 eV), respectively [38]. The N-6 can offer additional pseudo-capacitance, while the N-Q can enhance the conductivity [39]. Hence, the two N-containing functional groups are conducive to promoting the electrochemical performances.

The SEM and TEM images of as-obtained samples are displayed in Figure 4. Obviously, only with the carbonized treatment, the C sample mainly displays an irregular bulky structure and some debris, as well as a bit of pits exsiting on the surface (Figure 4a). On the basis of the reaction mechanism, the C element in material will react with KOH during the activation process to release CO_2_ gas [40]. Therefore, plentiful porosity will emerge in LSPCs samples (Figure 4b–d). After treating with an anneal temperature of 700 °C, the LSPC700 sample presents the numerous and uniform pores on the surface. In addition, with the relatively low temperature, the activation reaction is weak, and some areas of the LSPC700 sample still retain some similar irregular bulky structure and debris as the C sample. When the temperature increases up to 800 °C, as shown in Figure 4c, the LSPC800 sample possesses more abounding and wider pores compared with LSPC700. Besides, owing to the enhancement of activation reaction, a large number of pores emerge inside the sample and are connected with the pores on the surface to form an interconnected 3-dimensional structure. The interconnected 3-dimensional porous structure is beneficial to the rapid transport of ions, while the linked carbon wall is also conducive to improving the conductivity of material. Similarly, the LSPC900 also possesses an interconnected 3-dimensional structure, which contains numerous and uniform pores (Figure 4c). However, it is obvious that the carbon wall of LSPC900 becomes thin and even collapsed compared with LSPC800, which is mainly ascribed to the further evolving of pores by increasing the activation temperature [41]. The detail of morphology and the 3-dimensional porous structure of LSPC800 are further studied by TEM analysis, which are shown in Figure 4e,f. As displayed in Figure 4e, LSPC800 consists of numerous pores. These pores with sizes of a few nanometer were together to form a 3-dimensional porous structure. Besides, it can be seen in Figure 4f that amounts of intricate fringes demonstrate an amorphous carbon structure, which is in agreement with XRD and Raman measurements.

Figure 5 presents isotherms and these homologous PSD of LSPCs. As exhibited in Figure 5a, isotherms of all LSPCs samples belong to the type I in term of IUPAC (International Union of Pure and Applied Chemistry) classification [42]. These isotherms express the rapid rise at the low relatively pressure (P/P_0_ < 0.05) and the platform exists in P/P_0_ = 0.2–0.9 with no hysteresis between the adsorption and desorption branch. Herein, the as-obtained samples possess a typical high porous structure with the existence of plenty of micropores [43]. Furthermore, it is interesting that the adsorption capacity increases first and then decreases with the increasee of activation temperature. It can be explained that, as activation temperature rises from 700 °C to 800 °C, the activation process become active, resulting in more pores, but when activation temperature rises from 800 °C to 900 °C, owing to the excess activation, some weak pore structure collapse causes the reduction of porosity [44]. The above explanation is completely consistent with the detailed parameters of LSPCs, which are demonstrated in Table 2. These as-prepared LSPCs display a large specific surface area (up to 3089 m^2^ g^−1^) and high porosity (up to 1.644 cm^3^ g^−1^). Such high total pore volume can provide plentiful active sites and sufficient storage accommodation, which facilitate ion and charge adsorption as well as the storage of electrolyte [45,46]. The SSA, total pore volume, and micropore volume all present the tendency of increasing first and then declining, which is in agreement with the results of N_2_ adsorption and desorption isotherms. However, the mesopore volume of LSPCs increases gradually as the temperature goes up; the main explanation is that, with the intensification of the reaction, the micropores in the material are continuously etched, thus forming more mesopores. In Figure 5b, as proven in Figure 5b, the pore diameters of all LSPCs are basically located in range smaller than 4.0 nm and most are concentrated from 1 to 2 nm, which further confirms these materials contain plentiful micropores and some mesopores. These micropores, acting as the storage space of electrolyte, are the key for the electrodes to acquire a high specific capacitance. In addition, the pore size of these samples increases gradually as the temperature increases, and numerous mesopores with a diameter of 2–4 nm are produced for LSPC900 compared with LSPC700. These mesopores, acting as a buffer channel, can reduce ion transport resistance, and are conducive to improving the capacitance rate performance of carbon material [47]. Thus, such moderate PSD can not only obtain a high specific capacitance, but also retain the excellent capacitance rate performance.

The electrical conductivity of LSPCs is measured at different pressures (Figure 6). It can be observed that LSPCs samples all exhibit high electrical conductivity in the pressure of 10 MPa (the preparation pressure of electrodes), which is 533.175, 561.936, and 550.098 S m^−1^, respectively. Besides, the obtained conductivity values are superior to those of other carbon materials, such as nitrogen and oxygen co-doped carbon (421.9 S m^−1^) [48], hierarchical ordered mesoporous carbons/graphene composite (~420.8 S m^−1^) [49], and so on, which can mainly be ascribed to the high nitrogen content of the LSPCs. It is worth noting that the electrical conductivity of LSPCs increases first and then decreases with the increase of nitrogen content, and LSPC800 displays the highest electrical conductivity under different pressures. According to reported literatures, oxygen and defect are negative to effecting the electrical conductivity carbon materials [50,51]. LSPC700 possesses the relatively high oxygen content (14.88%) and LSPC900 achieves the largest *I*_D_/*I*_G_ value (1.29), and the two factors may cause the decline of electrical conductivity. Hence, LSPC800 achieves the highest electrical conductivity, which is conducive to the charge rapid transfer and reducing the IR drop and obtaining excellent rate performance for electrodes, even at a high current density.

### 3.2. Electrochemical Behaviors of LSPCs in Three-Electrode System

The electrochemical performances of LSPCs electrodes are exhibted in Figure 7. Figure 7a displays CV performances of LSPCs at 5 mV s^−1^. These curves are rectangular, manifesting the typical double-layer capacitive behaviour. In addition, LSPC800 achieves the largest integrated curve area compared with LSPC700 and LSPC900, demonstrating that it possessed the highest specific capacitance. The highest specific capacitance can be mainly ascribed to the highest porosity and moderate N element content of LSPC800. The CV curves of LSPC800 at different scan rates ranging from 2 to 100 mV s^−1^ are further presented in Figure 7d. With the increase in scan rate, these curves still hold the rectangle-like shapes even at 100 mV s^−1^, proving the superior capacitance rate property, which can be ascribed to quick ions transport by the interconnected 3-dimensional porous structure and excellent conductivity by high N content. Besides, similar outcomes are obtained from the GCD experiment. Figure 7b depicts the GCD curves of different LSPCs at a current density of 1 A g^−1^. Obviously, the linear and symmetrical GCD curves confirm that LSPCs have outstanding reversibility and ideal charge and discharge efficiency. LSPC800 possesses the longest charging and discharging time, inferring the highest specific capacitance, which is in agreement with the conclusion acquired from Figure 7a. Figure 7e shows the detailed GCD curves at different current densities ranging from 0.5 to 50 A g^−1^. These curves are nearly standard isosceles triangular shape, suggesting excellent capacitive behaviors. Meanwhile, the IR drop in curves is also extremely imperceptible, which can be attributed to the interconnected 3-dimensional porous structure, high porosity, and moderate heteroatomic content.

Figure 7c gives the Nyquist plots for LSPCs electrodes. These curves consist of three parts, a low, medium, and high frequency region [52]. Firstly, the semicircle closely means the charge transfer resistance (Rct) [53]. Compared with LSPC-700 and LSPC-900, LSPC-800 possesses the smallest semicircle, indicating its low charge transfer resistance, which can be attributed to the excellent conductivity of LSPC800 by N-Q. In the medium frequency region, the region is related to the slash named as Warburg impedance (Wi) [54]. It is observed that these obtained electrodes present a 45° slash, showing the ideal electrochemical performance. Finally, a straight sloping line in the high frequency region stands for the ion diffusion resistance of electrolyte in electrode pores. It is also demonstrated that the angle of the straight line increases first and then decreases as a function of the activation temperature. This indicates that the LSPC800 electrode possesses the smallest ion diffusion resistance, which can be ascribed to the developed and interconnected 3-dimensional porous structure. In addition, the electrical equivalent circuit is depicted in the inset, where *R*s is the electrolyte resistance and Re denotes the intrinsic resistance of electrodes. In Figure 7f, the specific capacitance of LSPCs calculated based on Equation (1) is aquired at different current densities. The specific capacitance of LSPC800 is 359.19 F g^−1^ at 0.5 A g^−1^, and this value is vitally superior to those of the other two samples (319.64 F g^−1^ for LSPC700 and 250.99 F g^−1^ for LSPC900). This result is principally owing to the larger specific surface area and pore volume of LSPC800, making for enough storage space of electrolyte. Even when current density reaches 50 A g^−1^, LSPC800 also abides an excellent capacitance value of 286.91 F g^−1^, which is still higher than those of LSPC700 and LSPC900 (245.48 and 198.58 F g^−1^). Besides, as current density goes up from 0.5 to 50 A g^−1^, the capacitance rate of LSPC800 maintains 79.79%, which profits from the outstanding physical and chemical characteristics.

### 3.3. Electrochemical Behaviors of LSPCs in SCs

To judge the actual application of the LSPC800 electrode, the performances of symmetrical SCs were further investigated.

#### 3.3.1. LSPC800-Based Aqueous SCs

Figure 8a demonstrates CV curves of the assembled aqueous SC using LSPC800 as electrode in different voltages at 20 mV s^−1^. It can be seen that excellent rectangular curves are presented as the upper limit voltage increases from 1.0 to 1.6 V, demonstrating that SCs with 1 M Li_2_SO_4_ can run in voltage of 1.6 V. Figure 8b exhibits curves of aqueous SC at different scan rates ranging from 2 to 100 mV s^−1^, with a voltage window of 0–1.6 V. As shown, the curves show rectangular shapes, manifesting the fast charge/discharge performance. Even when the scan rate reaches 100 mV s^−1^, the curve still exhibits a quasi-rectangle without any deformation, implying the excellent transport of ions, which is entirely owing to its high porosity and interconnected 3-dimensional porous structure. No peak is observed in the curves because lithium-ions in electrolyte are vitally matching with the LSPC800 sample, which also further proves the ideal capacitive behaviour. Moreover, in Figure 8c, LSPC800 GCD curves appear to have symmetrical triangular profiles with no distortions. Meanwhile, the internal resistance (IR) drop of SC is also vitally small. Even up to a current density of 50 A g^−1^, the curve only shows a ~0.3 V IR drop, manifesting that it possesses outstanding conductivity. From Figure 8d, LSPC800 acquires impressive specific capacitance of 253.78 F g^−1^, and the 74.21% of initial capacitance is well reserved at 50 A g^−1^. The above consequence also proves the excellent rate performance of aqueous SC, which profit from the high porosity and N content of LSPC800. As depicted in Figure 8d, at 0.5 A g^−1^, coulombic efficiency (CE) is relatively small (93.30%). This can be ascribed to the incomplete discharge caused by some side reactions. However, with increase of current density, the electrical double layer is dominant during charging/discharging, thus CE of LSPC800 reaches 99.95%, which is attributed to the rapid ions transport by interconnected 3-dimensional porous structure.

The Nyquist plot is presented in Figure 8e. From the low frequency region, the nearly vertical line indicates the small ion diffusion resistance. In the high frequency region, the relatively small semicircle also demonstrates the ideal electrochemical performance. Finally, Figure 8f displays the Ragone plot. LSPC800 achieves a high energy density of 22.44 Wh kg^−1^ at 199.48 W kg^−1^, which also presents 10.62 Wh kg^−1^ at an ultrahigh power density of 15,930.38 W kg^−1^. Furthermore, its ultrahigh power density is superior or comparable to some other biomass-based carbons (in Table 3), such as the mixture of coconut shell and sewage sludge-derived carbon, pig skin-derived carbon, corn starch-derived carbon, lentinus edodes-derived carbon, bagasse wastes-derived carbon, and pea skin-derived carbon. In brief, high porosity and moderate heteroatoms of LSPC800-based electrode achieve ultrahigh power density.

#### 3.3.2. LSPC800-Based All-Solid-State SCs

The schema of assembled SC using PVA/KOH gel are displayed in Figure 9. It can be seen that the sandwich-like model comprises LSPC800 sample, solid electrolyte, as well as nickel foam. As depicted in Figure 10a, the voltage range of the all-solid-state SC is the same as aqueous KOH electrolyte SCs (0–1 V). Under the voltage window, as the scan rate increases up to 100 mV s^−1^, the curves of LSPC800-based all-solid-state SC still remain the typical rectangle-like outline with only a little deformation, which is because the solid electrolyte transports relatively slowly through the material. In Figure 10b, when current density increases to 20 A g^−1^, these GCD curves exhibit an isosceles triangle, which also displays the high charging and discharging efficiency and outstanding capacitive reversibility of LSPC800 electrode. Meanwhile, these vitally small IR drop are displayed. Even at 50 A g^−1^, an IR drop of 0.412 V indicates low internal resistance. As seen in Figure 10c, the assembled all-solid-state SC presents a superior capacitance value (312.64 F g^−1^), while it still maintains 227.40 F g^−1^ with 72.73% of capacitance rate at 50 A g^−1^, which demonstrates that all-solid-state SC also has an excellent rate capability. Figure 10c also plots the coulombic efficiency at various current densities. It can be observed that coulombic efficiency is still relatively high (92.56%) at a small current, and the coulombic efficiency has reached up to 98.23% with the current density of 10 A g^−1^, indicating there is basically no heat loss during the charge/discharge process [29]. The Nyquist plot in Figure 10d displays a very small semicircle, manifesting the high efficient charge transfer. What is more, the ESR value of all-solid-state SC is ~0.740 Ω, which is very low and consistent with the low IR drop. It is interesting that the ESR value of all-solid-state SC exceeds that of the aqueous SC device (~0.594 Ω), and the phenomenon is mainly because of slow diffusion of the solid electrolyte [59]. The two important parameters are energy density and power density, which are illustrated in Figure 10e. The all-solid-state SC presents the largest energy density of 10.74 Wh kg^−1^ with a power density of 124.34 W kg^−1^ at a current density of 0.5 A g^−1^. The energy density overtops some previous literatures, including porous carbon (6.5 Wh kg^−1^) [60], carbon nanofibers (5.9 Wh kg^−1^) [61], and B-doped graphene material (5.3 Wh kg^−1^) [62]. When current density goes up to 50 A g^−1^, the all-solid-state SC still remains at 2.58 Wh kg^−1^ with an impressive power density of 7146.13 W kg^−1^. To evaluate the cycling stability of the all-solid-state SC, the 10,000 GCD consecutive runs process was carried out at a current density of 5 A g^−1^, which is displayed in Figure 10f. The capacitance retention of the SC device is ultrahigh (93.64%), demonstrating splendid stability.

## 4. Conclusions

To sum up, 3-dimensional porous carbon with a high N content carbon was triumphantly fabricated using biomass precursor by simple carbonization as well as KOH activation approaches. These prepared longan shell 3-dimensional porous carbons (denoted as LSPCs) possess large SSA (maximum of 3089 m^2^ g^−1^) and abundant porosity (maximum of 1.644 m^3^ g^−1^), which make for improving the specific capacitance by storing more electrolyte ions. The optimized material that is named after LSPC800 not only has the above advantages, but also achieves the high nitrogen content (1.14%), which is conducive to introducing faradaic pseudocapacitance and enhancing the conductivity. LSPC800 displays an excellent capacitance (359 F g^−1^) in the three-electrode system. Besides, superior performances are obtained though assembling SCs in aqueous (1 M Li_2_SO_4_) and solid (KOH/PVA gel) electrolyte. On the one hand, the aqueous SC acquires the excellent specific capacitance of 254 F g^−1^ with a high voltage window (0–1.6 V) and an ultrahigh power density of 15,930.38 W kg^−1^. On the other hand, these assembled SCs with gel electrolyte also achieve a high capacitance value of 313 F g^−1^, and its capacitance retention is 93.64% at 5 A g^−1^ at the 10,000th cycle. The energy density of all-solid-state SC (10.74 Wh kg^−1^) is compared with the reported results in the references. It is obvious that the value is also satisfactory. This study demonstrates that the 3-dimensional porous carbon can be facile fabricated using biomass material by simple carbonization and activation, as well as further opening up a way of improving the value-added utilization of biomass materials for high-performance SCs application.

## Figures and Tables

**Figure 1 nanomaterials-10-00808-f001:**
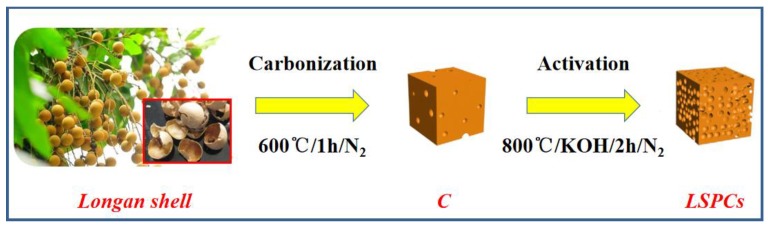
Schematic illustration of turning longan shell into longan shell 3-dimensional porous carbons (LSPCs). KOH, potassium hydroxide.

**Figure 2 nanomaterials-10-00808-f002:**
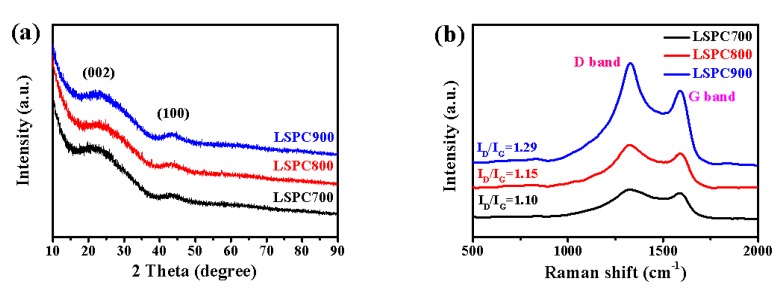
(**a**) X-ray diffraction (XRD) pattern and (**b**) Raman spectra of LSPCs.

**Figure 3 nanomaterials-10-00808-f003:**
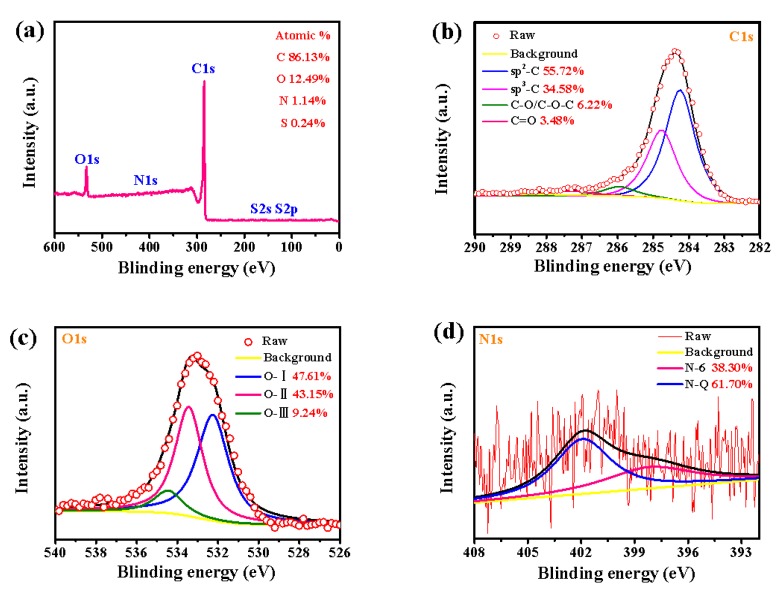
XPS spectra of of LSPC800: (**a**) full spectrum, (**b**) C1s, (**c**) O1s, and (**d**) N1s.

**Figure 4 nanomaterials-10-00808-f004:**
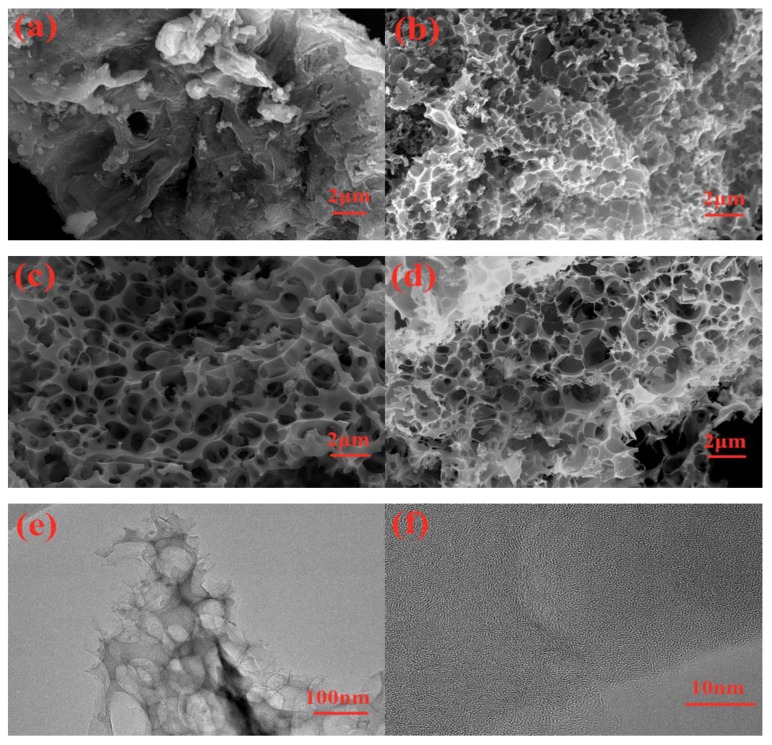
Scanning electron microscope (SEM) images of (**a**) C, (**b**) LSPC700, (**c**) LSPC800, and (**d**) LSPC900 at the same magnification; and (**e**) transmission electron microscope (TEM) and (**f**) HTEM (High voltage transmission electron microscope) images of LSPC800.

**Figure 5 nanomaterials-10-00808-f005:**
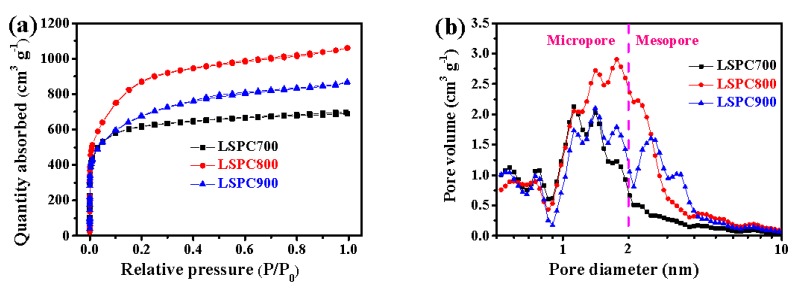
(**a**) N_2_ adsorption and desorption isotherms and (**b**) pore size distribution of the LSPCs.

**Figure 6 nanomaterials-10-00808-f006:**
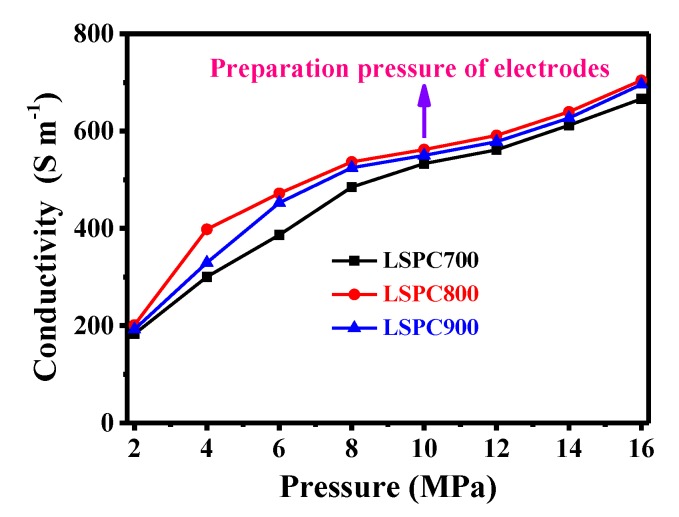
The electrical conductivity of LSPCs under the pressure ranging 2 to 16 MPa.

**Figure 7 nanomaterials-10-00808-f007:**
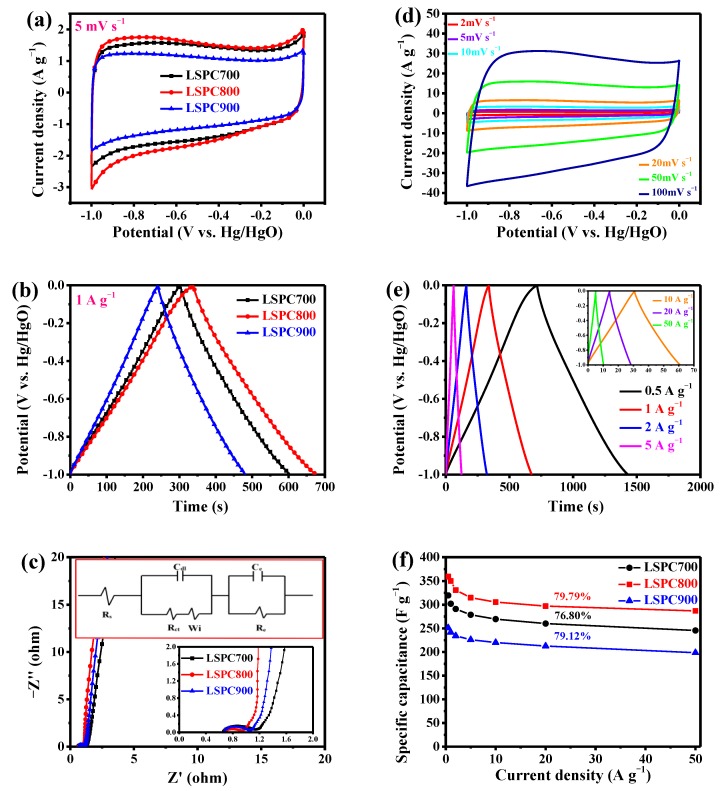
Electrochemical performances of the LSPCs in 6 M KOH using a three-electrode system: (**a**) cyclic voltammetry (CV) curves at 5 mV s^−1^; (**b**) galvanostatic charge–discharge (GCD) curves at a current density of 1 A g^−1^; (**c**) the Nyquist plots and the equivalent circuit in the inset; (**d**) CV curves of LSPC800 at scan rates varying from 5 to 100 mV s^−1^; (**e**) GCD curves of LSPC800 at current densities varying from 0.5 to 50 A g^−1^; and (**f**) specific capacitance at different current densities.

**Figure 8 nanomaterials-10-00808-f008:**
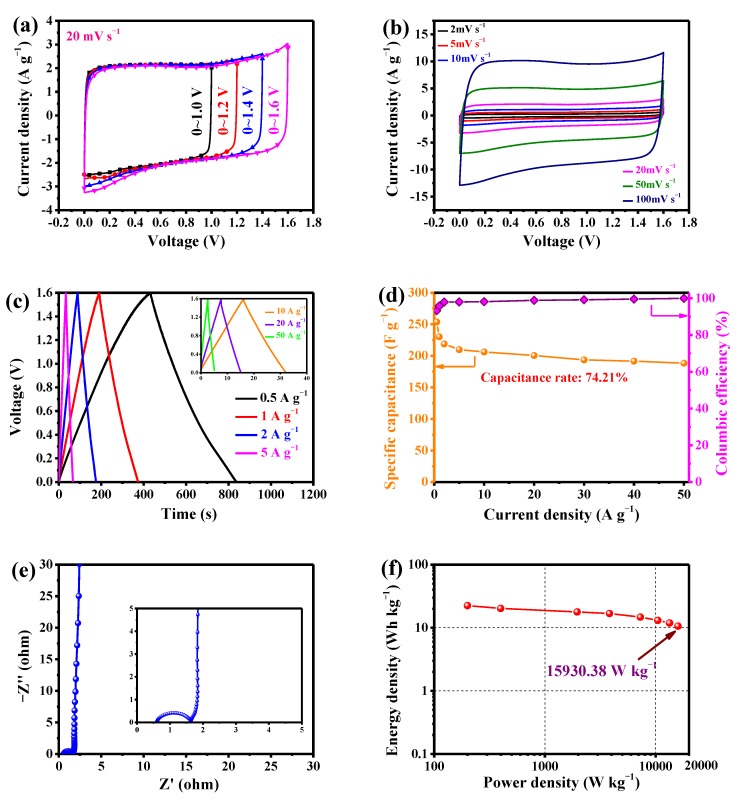
Electrochemical performances of the as-assembled LSPC800-based supercapacitor (SC) device in 1 M Li_2_SO_4_ electrolyte: (**a**) CV curves upon different voltage windows at 20 mV s^−1^; (**b**) CV curves at scan rates varying from 5 to 100 mV s^−1^; (**c**) GCD curves at different current densities; (**d**) specific capacitance and coulombic efficiency of the SC device at different current densities; (**e**) the Nyquist plot of the assembled SC device; and (**f**) the Ragone plot of the assembled SC device.

**Figure 9 nanomaterials-10-00808-f009:**
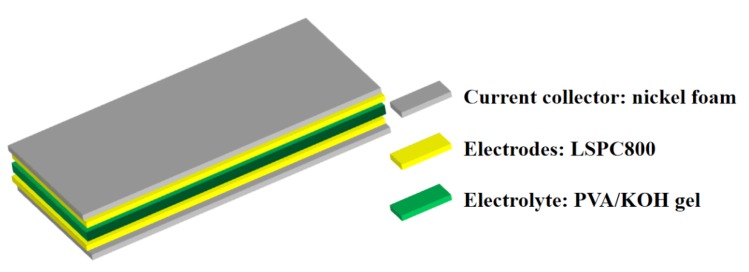
Schematic illustration of the all-solid-state SC device. PVA, polyvinyl alcohol.

**Figure 10 nanomaterials-10-00808-f010:**
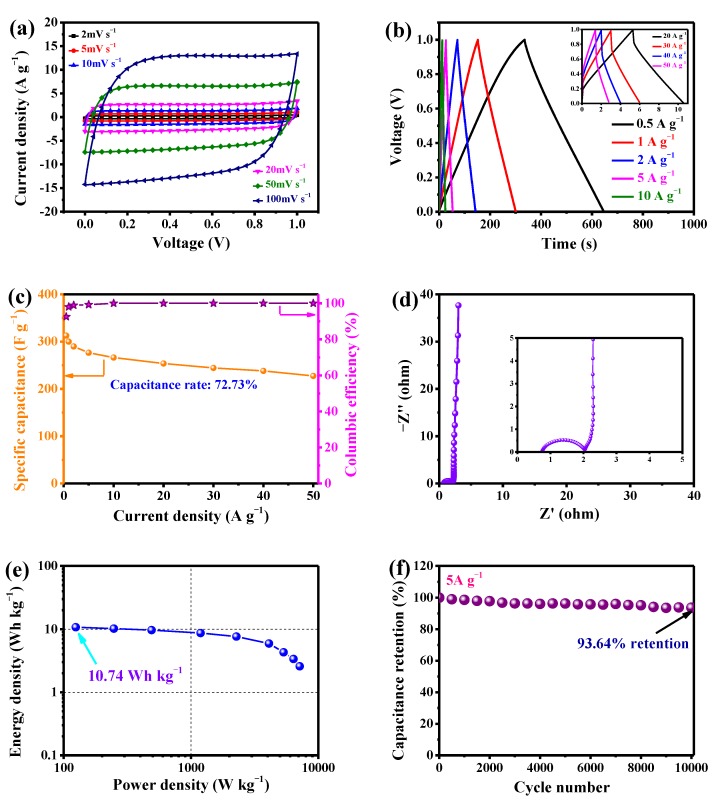
Electrochemical performances of the assembled all-solid-state SC device using LSPC800 as electrode in PVA/KOH gel electrolyte: (**a**) CV curves at scan rates varying from 5 to 100 mV s^−1^; (**b**) GCD curves at different current densities; (**c**) specific capacitance and coulombic efficiency at different current densities; (**d**) the Nyquist plot; (**e**) the Ragone plot; and (**f**) the long-cycles performance at 5 A g^−1^.

**Table 1 nanomaterials-10-00808-t001:** Element content (atomic %) of the longan shell 3-dimensional porous carbons (LSPCs) by XPS.

Sample	C (%)	O (%)	N (%)	S (%)
LSPC700	83.52	14.88	1.32	0.28
LSPC800	86.13	12.49	1.14	0.24
LSPC900	90.31	8.65	0.86	0.18

**Table 2 nanomaterials-10-00808-t002:** Pore structure parameters of the LSPCs.

Sample	S_BET_ (m^2^ g^−1^) ^a^	V_t_ (cm^3^ g^−1^) ^b^	V_mic_ (cm^3^ g^−1^) ^c^	V_mes_ (cm^3^ g^−1^) ^d^	V_mes_/V_t_ (%)
LSPC700	2299	1.071	0.827	0.244	22.78
LSPC800	3089	1.644	1.069	0.575	34.29
LSPC900	2401	1.346	0.713	0.633	47.03

^a^ Specific surface area, obtained by BET mode; ^b^ total pore volume, analyzed at P/P_0_ = 0.99; ^c^ micropore volume, estimated from the t-plot method; ^d^ mesopore volume, calculated by the difference between total pore and micropore volume.

**Table 3 nanomaterials-10-00808-t003:** Comparison of the maximum power density of MAC-800 with other biomass porous carbon electrodes.

No.	Electrode Material	Electrolyte	Voltage Window	Maximum Power Density (W kg^−1^)	Ref.
1	Hierarchical porous carbons from the mixed biomass wastes	1 M Na_2_SO_4_	0–1.8 V	9000	[38]
2	Pig skin-derived porous carbon	EMIMBF_4_	0–3.5 V	8750.6	[55]
3	3 D porous carbon from corn starch	2 M [BMIm]BF_4_/AN	0–2.6 V	~7000	[19]
4	Hierarchical porous carbon from waste lentinus edodes	1 M H_2_SO_4_	0–1.0 V	5000	[56]
5	Hierarchical structured carbon derived from bagasse wastes	6 M KOH	0–1.0 V	10,673	[57]
6	Activated carbon derived from pea skin	1M LiClO_4_ in EC/PC	0–2.0 V	17,900	[58]
7	Longan shell 3-dimensional porous carbon	1 M Li_2_SO_4_	0–1.6 V	15,930.38	**This work**

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
