# Peer review of "3-Dimensional Porous Carbon with High Nitrogen Content Obtained from Longan Shell and Its Excellent Performance for Aqueous and All-Solid-State Supercapacitors"

_nanomaterials, 2020, doi:10.3390/nano10040808_

Round 1
Reviewer 1 Report
In this study, authors report the production of carbon materials from longan shell by simple carbonization and then the interconnected 3-dimensional porous structure was obtained via KOH activation treatment. This work demonstrates its application as active material in aqueous and all-solid-state supercapacitor. In terms of the electrochemical performance, as-prepared porous carbon from longan exhibits high electrical performance. I recommend to publish this manuscript in Nanomaterial after minor changes.
- Authors claimed that the carbon obtained from longan bio-source is rich in nitrogen, leading the its high electrical conductivity. The electrical conductivity results should be added in the manuscript to ensure N effect.
- Please check Figure numbering in the text (ig. 4 and 6)
Reviewer 2 Report
Liu et al have synthesized nitrogen doped carbon from longan shell. I highly appreciate the authors for enriching the manuscript with lot of data. The data presentation and the results section is well discussed. I highly recommend this article to be published in Nanomaterials journal without any further modifications.
Author Response
Thank you for your approval and recommendation of our work.
Reviewer 3 Report
The manuscript is about supercapacitor development from 3-dimensional porous carbon with high nitrogen content obtained from longan shell. The manuscript is clearly written and well performed experimental part. The manuscript can be published in Nanomaterials.
Author Response

(The authors gave the same response as above.)
